# Change in Bone Mineral Density in Stroke Patients with Osteoporosis or Osteopenia

**DOI:** 10.3390/ijerph19158954

**Published:** 2022-07-23

**Authors:** Do-Hee Lee, Min-Cheol Joo

**Affiliations:** Department of Rehabilitation Medicine, Institute of Wonkwang Medical Science, Wonkwang University School of Medicine, 895, Muwang-ro, Iksan 54538, Korea; dlehgml833@wkuh.org

**Keywords:** stroke, osteoporosis, osteopenia, bone mineral density

## Abstract

We aimed to investigate the correlation between changes in bone mineral density (BMD) of the lumbar spine (LS) and femoral neck (FN) and osteoporosis-related factors in stroke patients with osteoporosis or osteopenia, and we suggest the need for active rehabilitation treatment. This study included 63 osteoporosis and 34 osteopenia patients who underwent a BMD test following primary stroke onset. The osteoporosis group was followed up with a BMD test after 12 months of bisphosphonate treatment, and the osteopenia group was followed up without medication. The correlation between BMD changes and functional factors was analyzed, biochemical markers were measured, and hematology tests were performed. In the osteoporosis group, a significant increase was observed in LS BMD (*p* < 0.05), and in the osteopenia group, there was a significant decrease in FN BMD (*p* < 0.05). The group with a functional ambulatory category of 1 or more showed a significant improvement in BMD (*p* < 0.05). Comparative analysis was performed on various indicators, but no significant correlation was found between any variable. In stroke patients with osteoporosis or osteopenia, early appropriate drug treatment is important to prevent bone loss and reduce the risk of fractures, and comprehensive rehabilitation treatment, such as appropriate education and training to prevent falls, is essential.

## 1. Introduction

Osteoporosis is a disease in which bone strength declines and the risk of fractures increases [1]. According to the World Health Organization, a bone mineral density (BMD) T-score of −2.5 or below is a defining feature of osteoporosis. Stroke patients are known to have developed osteoporosis for various reasons, such as limited exercise and lack of nutrients [2]. Bone loss begins immediately after stroke, continues until 3–4 months after onset, and progresses at a slower rate until one year after onset [3]. Stroke patients fall frequently due to paralysis and loss of balance, and as a result, femur fractures are 2–4 times more common [4,5]. Most fractures after stroke occur in the hemiplegic side of the body because the BMD in this side decreases by 4.6–14% compared with the unaffected side [6]. Fractures due to osteoporosis in stroke patients are difficult to treat and cause various complications, which can lead to death in severe cases. Furthermore, fractures prevent proper rehabilitation treatment, and there are many social and economic burdens, such as increased distress for patients and their guardians, increased medical costs, extended hospitalization periods, and the need for long-term rehabilitation treatment [7]. Therefore, prevention of fractures due to osteoporosis after stroke is crucial for comprehensive and intensive rehabilitation treatment in early stroke patients. Treatment of osteoporosis in stroke patients includes various osteoporosis treatment drugs, strength training, weight-bearing training, gait training, and education for fall prevention [8]. Bisphosphonate, the most widely used drug for osteoporosis, is also known to be effective in the treatment of osteoporosis accompanying stroke [9].

Along with osteoporosis, many stoke patients are also diagnosed with osteopenia. Osteopenia is defined by a BMD T-score of −1 to −2.5. The presence of osteopenia increases the likelihood of osteoporosis, which increases the risk of fractures [10]. Nevertheless, studies on changes in BMD in stroke patients with osteopenia are limited, and there are very few studies on the appropriate treatment and management of osteopenia. In addition, there are few studies comparing BMD changes between stroke patients with osteoporosis and stroke patients with osteopenia.

Thus, we designed a prospective study to compare BMD changes in hemiplegic stroke patients with osteoporosis treated with bisphosphonates to those in osteopenia patients that received no drug treatment at a 1-year follow-up visit that assessed the need for active drug treatment. In addition, by analyzing the correlation between osteoporosis-related factors and changes in BMD, we recommend comprehensive rehabilitation, including prevention of fractures, for stroke patients.

## 2. Materials and Methods

### 2.1. Subjects

The subjects included patients who were treated for a stroke at the Department of Rehabilitation Medicine at Wonkwang University Hospital from 2011 to 2019. The inclusion criteria were: (1) patients diagnosed with primary cerebral infarction or cerebral hemorrhage; (2) patients diagnosed with osteoporosis or osteopenia according to BMD; and (3) patients who underwent hematologic examination and functional evaluation. The exclusion criteria were: (1) patients with brain injury caused by trauma; (2) patients with comorbidities that could affect BMD, such as a history of spine or femur fracture, or a metabolic disease that may affect bone metabolism (including liver, renal, or thyroid disease); and (3) patients taking drugs that affect bone metabolism.

All subjects provided informed consent for inclusion before they participated in the study. The study was conducted in accordance with the Declaration of Helsinki, and the protocol was approved by the Institutional Bioethics Committee of Wonkwang University Hospital (IRB Number: 2018-02-034).

### 2.2. Methods

We evaluated each patient’s demographic characteristics, stroke type, and duration of disease. At the start of the study (T0), BMD and hematology tests were conducted in addition to evaluation of walking ability and functional level of daily activities. At the end of the study (T1), after a period of 12–14 months, the BMD and hematology tests were conducted again using the same device. We administered zoledronic acid, a type of bisphosphonate, to osteoporotic patients with a BMD T-score of −2.5 or below. In addition, the occurrence of adverse effects, such as bisphosphonate-related osteonecrosis of the jaw (BRONJ) [11,12], atypical femoral bone fracture [13], nonspecific conjunctivitis, atrial fibrillation, and flu-like symptoms (fever, fatigue, and myalgia) [14,15], was evaluated during the administration period.

#### 2.2.1. BMD Test

BMD was measured in the lumbar spine (LS) and hemiplegic femoral neck (FN) using dual-energy X-ray absorptiometry (GE Lunar Corporation, Madison, WI, USA). The tests were performed at 8:00 in the morning before consumption of food or water, and excessive calcium intake was prohibited before testing so as not to affect the values. The standard deviation (T-score) and absolute BMD (g/cm^2^) were obtained from the average bone mass of young adults, and results from before and after treatment were compared. A T-score of −1.0 to −2.5 was classified as osteopenia, while a T-score of −2.5 or below was classified as osteoporosis.

#### 2.2.2. Biochemical Markers and Hematology Tests

Hematology tests measured the levels of osteocalcin (OC), a bone formation marker; C-telopeptide of collagen type 1 (CTX), a bone resorption marker; and 1,25-dihydroxyvitamin D (1,25-(OH)2D), an active form of vitamin D. Samples were collected before meals and within 2 h of waking up in order to reduce the error of transient changes. A blood vitamin D concentration of 10 ng/mL or less was defined as deficient, 11–29 ng/mL as insufficient, and 30 ng/mL or above as sufficient, and the degree of improvement in BMD was evaluated for each group [16].

#### 2.2.3. Functional Evaluation

The functional ambulatory category (FAC) was used to evaluate the subject’s walking ability [17], and subjects were then divided into two groups based on their FAC: the FAC 0 group included subjects who were incapable of walking, and the FAC ≥ 1 group included subjects who required physical assistance when moving. Modified Barthel index (MBI) was used to measure disability or dependence in activities of daily living (ADL) [18]. The subjects were divided into two groups: those with an MBI ≤ 32, which included patients who required full assistance in performing daily activities, and those with an MBI > 32. A manual muscle test (MMT) was conducted to evaluate the strength of the hip extensor and knee extensor [19,20]. One group showed MMT grades of F and above for both the hip and knee extensors, while members of the other group showed an MMT grade of F or below for either the hip extensor or knee extensor.

#### 2.2.4. Statistical Analysis

Statistical significance was defined as a *p*-value of <0.05. All statistical analyses were performed using SPSS ver. 26.0 software package (IBMSPSS, Armonk, NY, USA). A paired *t*-test was used to compare absolute BMD and hematologic values within each group at the beginning and 1-year mark, and an independent *t*-test was used to compare changes in absolute BMD between groups. Pearson’s correlation analysis was also performed to determine whether there were correlations between BMD and several indicators, such as degree of vitamin D deficiency.

## 3. Results

### 3.1. Demographic Characteristics

The total number of subjects was 97, with a gender distribution of 9 males and 88 females. The average age was 73.2 ± 9.9 years. There were 63 subjects diagnosed with osteoporosis and 34 subjects diagnosed with osteopenia. The average period from the onset of stroke to BMD-decline was 4.5 ± 3.7 months. In terms of stroke type, cerebral infarction had occurred in 73 patients and cerebral hemorrhage in 24 patients. The division of subjects according to strength and functional level is shown in Table 1, along with the degree of vitamin deficiency of each group.

### 3.2. Changes in BMD

As a result of comparing the absolute BMD (g/cm^2^) before (T0) and after (T1) treatment with bisphosphonates for 12 months in patients diagnosed with osteoporosis, LS BMD showed a statistically significant increase from 0.667 ± 0.141 to 0.722 ± 0.141( *p* < 0.05), but FN BMD showed no statistically significant change (T0: 0.524 ± 0.073; T1: 0.526 ± 0.081; *p* > 0.05). In patients diagnosed with osteopenia, there was no statistically significant difference in LS BMD after 12 months (T0: 0.855 ± 0.053; T1: 0.865 ± 0.061; *p* > 0.05), but FN BMD showed a statistically significant decrease from 0.674 ± 0.091 to 0.615 ± 0.057 (*p* < 0.05) (Table 2).

### 3.3. Correlation between Functional Evaluation and Changes in BMD

When comparing the FAC 0 and the FAC ≥ 1 in the total group, the changes in the absolute BMD of the LS were −0.004 ± 0.114 and 0.046 ± 0.069, respectively, and the changes in the BMD of the FN were −0.065 ± 0.045 and −0.024 ± 0.056, respectively, showing a statistically significant improvement in the FAC ≥ 1 group (*p* < 0.05). Even when comparing the osteoporosis and osteopenia groups, there was no statistically significant difference, while the FAC ≥ 1 group showed more improvement in BMD in the LS and FN (Figure 1). There was no statistically significant difference in the correlation between the change in BMD with MMT or MBI (Figure 2).

### 3.4. Correlation between BMD and Clinical Variables in Stroke Patients

Comparative analysis was performed on each groups’ degree of vitamin D deficiency and BMD level, but there was no statistically significant correlation between each variable (Table 3).

### 3.5. Changes in Biochemical Markers and Hematology Tests

In the osteoporosis group, the OC level showed a statistically significant decrease from 13.2 ± 5.5 at T0 to 11.36 ± 5.1 at T1 (*p* < 0.05), and the CTX level also showed a statistically significant decrease from 0.3 ± 0.2 at T0 to 0.21 ± 0.1 at T1 (*p* < 0.05). In the osteopenia group, the OC and CTX levels at T0 and T1 were not statistically different. The total vitamin D levels at T0 and T1 were not statistically different in either group (Table 4).

### 3.6. Adverse Effects

During the one-year period of follow-up, adverse effects were observed in 6 out of 63 patients (9.5%) in the treatment group, and the types of adverse effects were three cases of fever, one of chills, and two of myalgia. The severity of all side effects was mild, and most of them subsided within 2–3 days after follow-up. There were no serious adverse effects such as death, BRONJ, or atypical femoral bone fracture in the rest of the patient group.

## 4. Discussion

There are various causes of osteoporosis in stroke patients, including exercise restriction and weight-bearing reduction due to paralysis, insufficient nutrient intake due to eating disorders, intake of various medications, and vitamin D decrease due to insufficient sunlight [2]. Changes in BMD after stroke have reportedly been correlated with age, degree of paralysis, duration of paralysis, blood calcium level, vitamin D concentration, and vitamin K concentration during the first year, and degree of paralysis and blood vitamin D concentrations after two years [21]. The risk of fractures increases after a stroke; in particular, FN fracture occurrence is 2–4 times higher in stroke patients compared with the general population of the same age [4,5]. As such, stroke patients are more likely to develop osteoporosis, which increases the incidence of spine or FN fractures, which in turn increases the length of hospital stay and medical expenses. Therefore, early detection and management of osteoporosis are crucial [7]. Various methods are known for the treatment of osteoporosis, including behavioral changes, diet, and drug treatment [8]. Bisphosphonates, one of the drug treatment options, are widely used, and their positive effects on stroke patients have been reported in previous studies [9]. Thus, this study investigated the change in BMD of the LS and FN after administration of bisphosphonates in stroke patients with osteoporosis and in stroke patients with osteopenia at a 1-year follow-up examination to assess the need for active rehabilitation treatment and management. A statistically significant increase was found in LS BMD but not in FN BMD after 1 year of bisphosphonate administration in stroke patients with osteoporosis. FDA research and other studies have shown that when using bisphosphonate for more than 5 years, LS BMD is further increased and FN BMD is maintained [22]. It is therefore evident that bisphosphonate treatment is more effective in the LS. However, as the patients in the present study are stroke victims, the pattern of BMD change according to treatment may differ from that of the general population. To understand the above results, since the weight load does not appear on the hemiplegic side in stroke patients, one must consider that FN bone loss is more common on the hemiplegic side than on the unaffected side [6]. A previous study also reported that when comparing the BMD changes over time in stroke patients, there was no significant difference in LS BMD, but that FN BMD decreased further by 5.2% on the hemiplegic side and by 2.1% on the non-hemiplegic side [23]. Since FN BMD in the osteoporosis group of this study was measured on the hemiplegic side, bone loss was expected to appear over time. Instead, bisphosphonate treatment was able to prevent bone loss, and it also improved LS BMD. This study found that bisphosphonate treatment in the osteoporosis group was helpful in reducing the incidence of spine fractures after stroke by improving LS BMD. In fact, according to one study, the risk of spine fractures was reduced by 35–50%, with a 1–6% improvement in LS BMD after the administration of bisphosphonates [24]. Considering the existing literature [25] stating that the risk of stroke increases with reduced physical activity due to osteoporotic spine fractures, it follows that osteoporosis treatment is effective in preventing the recurrence of stroke to some extent. In addition, this study showed that the use of bisphosphonates in the osteoporosis group could prevent a marked decrease in FN BMD compared with the osteopenia group. These findings should be considered, as early appropriate drug treatment is important to prevent bone loss and to reduce the risk of fractures in osteoporosis patients overall, and comprehensive rehabilitation treatment, such as appropriate education and training to prevent falls, is essential.

The results of this study show a statistically significant decrease in FN BMD at 1-year follow-up of the patients in the osteopenia group who were not treated with osteoporosis medications. Osteoporosis medications such as bisphosphonates were not administered to patients with osteopenia because in Korea, health insurance coverage of osteoporosis medications is only possible for patients with osteoporosis marked by a T-score of −2.5 or below on a BMD test, or with osteoporotic fractures confirmed by radiographs. For this reason, in the osteopenia group, hemiplegic-side FN BMD was decreased as with the general stroke patients. LS BMD also changed over time, but seemingly to a lesser degree than FN BMD. Considering the increased risk of fall due to decreased muscle strength and balance after stroke, in patients with osteopenia, this significant decrease in FN BMD can lead to femur fracture resulting from a fall. Recent studies have shown that bisphosphonate treatment in osteopenia women has beneficial effects, such as increased BMD and reduced fracture rate [26], so medication is also likely to be effective for the osteopenia group. Therefore, even in patients diagnosed with osteopenia after stroke, early active drug treatment and dietary management, such as calcium and vitamin D intake, are considered necessary.

The relationship between functional levels, such as muscle strength, balance ability, and walking ability, after stroke and changes in BMD has not been clearly established. In a previous study, it was reported that the asymmetrical weight bearing on the hemiplegic side during standing or walking promoted bone loss, so the BMD of the hemiplegic side is further decreased [27]. In addition, among the various previous studies on this topic, one reported that BMD is not related to muscle strength, spasticity, or the ability to perform activities of daily living [28], while another study reported that BMD is correlated with walking ability and that bone loss can be reduced when standing and gait training are started within 2 months of onset [29]. The results of the present study show no statistically significant correlation between daily living performance and muscle strength and the changes in BMD in stroke patients. However, when comparing the group with FAC 0 and the group with FAC ≥ 1 to assess gait ability, the group with FAC ≥ 1 showed a statistically significant improvement in the degree of BMD improvement in the LS and FN. This is similar to the results of a previous study that found a significantly smaller decrease in BMD in the ambulatory group when comparing FN BMD in a wheelchair group at 1-year follow-up to that in an ambulatory group [30]. Another study that showed that people who can walk independently tend to have a lesser degree of BMD loss compared to those who cannot [31]. These results indicate that the lack of muscle control that persists after stroke may be one of the factors that affect BMD, but the accompanying weight bearing is another important determinant of BMD. Since an increase in bone mass is associated with bone loading, weight-bearing activities of daily living, such as walking, are very important for maintaining an individual’s FN BMD [32]. It is thought that the group with FAC ≥ 1 showed improvement in BMD levels in this study through a more active participation in rehabilitation training and weight-bearing exercises during hospitalization. Based on this, it can be assumed that the gait ability measured by FAC is a determinant of bone loss in the first year after stroke. This suggests the necessity of preventing muscle loss and reducing the risk of bone loss and fractures through comprehensive and active rehabilitation treatment, such as weight-bearing training, gait training, and fall prevention, in the early stages.

Among laboratory tests, OC is a marker related to bone formation, and CTX is a marker related to bone resorption. Bone resorption is known to increase after stroke [33]. However, it is known that when bisphosphonates are used, both bone resorption and bone formation markers decrease due to the inhibition of the bone-remodeling process [34]. This study also found that OC and CTX significantly decreased after 1 year in the osteoporosis group treated with bisphosphonates. Regarding the correlation between serum OC, CTX, and BMD, results vary from study to study. In one study, a negative correlation was found between serum OC, CTX, and the BMD of the FN [35], but in another study, no correlation was found between serum OC and the LS or FN BMD [36]. In this study, there was no significant correlation between the changes in BMD and serum OC and CTX. The reason for these varying results may be that there are individual differences in several factors such as gender, age, and mobility, when evaluating the serum level, and it is thought that consistent results are difficult to obtain due to the variability of the analysis [37].

Blood vitamin D levels are often deficient following stroke [38]. However, the relationship between vitamin D deficiency and the prevalence of osteoporosis in stroke patients is not yet clear. In one study, vitamin D deficiency was found in 71% of stroke patients, although neither the presence nor absence of vitamin D deficiency was confirmed to be significantly correlated with the prevalence of osteoporosis [39]. This study also showed a high prevalence of insufficient or deficient vitamin D levels in 73.2% of the patients, but there was no significant difference between the osteoporosis and osteopenia groups at 71.5% and 76.5%, respectively. Furthermore, comparing the correlation between the level of vitamin D deficiency and BMD did not yield any significant results. However, given the high prevalence of vitamin D deficiency in stroke patients and the associated risk of fracture [40], vitamin D supplementation is considered necessary in stroke patients by periodically checking vitamin D levels.

A limitation in this study is that since it is based on a specific Korean group, the results are less generalizable. Additionally, the average age of the target patient group was 70 years or older, most of them were women, so the effects of postmenopausal osteoporosis and senile osteoporosis could not be completely excluded. In the future, it will be necessary to evaluate BMD in various areas, such as the humerus, where bone loss occurs frequently.

## 5. Conclusions

In stroke patients with osteoporosis, the administration of osteoporosis drugs that suppress bone absorption produced a significant improvement in LS BMD but no significant change in FN BMD. In follow-up examinations of osteopenia patients, the BMD of the FN showed a significant decrease. Especially in patients with impaired gait ability, it is important to conduct early and active rehabilitation treatment, such as weight bearing and gait training. In stroke patients with osteoporosis or osteopenia, early appropriate drug treatment is important to prevent bone loss and reduce the risk of fractures, and comprehensive rehabilitation treatment, such as appropriate education and training to prevent falls, is essential.

## Figures and Tables

**Figure 1 ijerph-19-08954-f001:**
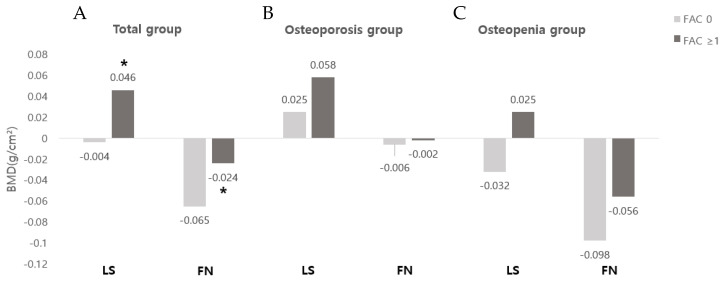
Changes in absolute BMD according to FAC: (**A**) total group; (**B**) osteoporosis group; (**C**) osteopenia group. Abbreviations: LS, lumbar spine; FN, femoral neck; FAC, functional ambulatory category; * *p* < 0.05 was analyzed by independent *t*-test.

**Figure 2 ijerph-19-08954-f002:**
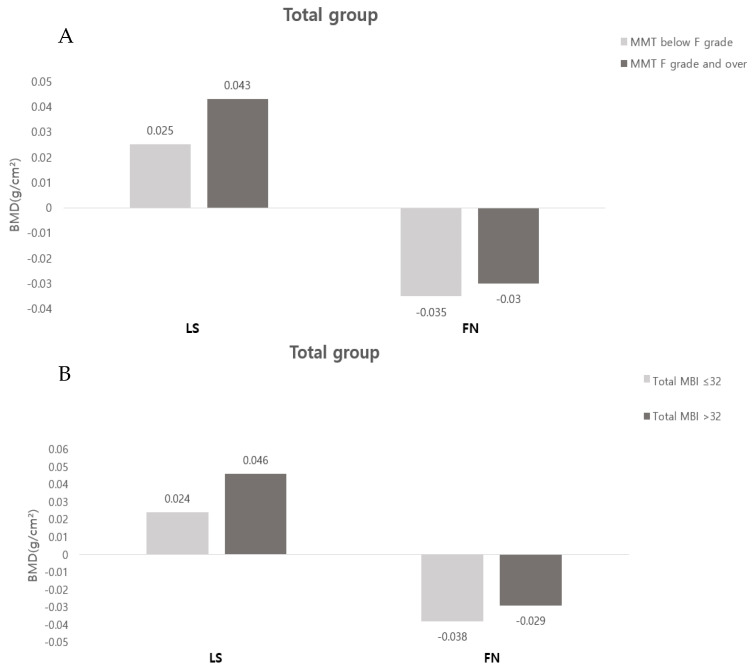
Changes in absolute BMD according to MMT and MBI: (**A**) comparison according to MMT; (**B**) comparison according to MBI. Abbreviations: LS, lumbar spine; FN, femoral neck; MMT, manual muscle test; MBI, modified barthel index.

**Table 1 ijerph-19-08954-t001:** General characteristics of the subjects.

Parameters		Osteoporosis (n = 63)	Osteopenia (n = 34)
Mean age (year)		73.6 ± 9.6	72.5 ± 9.4
Duration (month)		4.3 ± 3.5	4.7 ± 3.7
Sex	Male	5	4
	Female	58	30
Type of stroke	Infarction	44	29
	Hemorrhage	19	5
MMT of hip and knee	Below F grade	17	8
	F grade and over	46	26
MBI (total)	≤32	20	9
	>32	43	25
FAC	0	15	6
	≥1	48	28
Vitamin D	Normal (≥30 ng/mL)	18	8
	Insufficiency (11–29 ng/mL)	39	23
	Deficiency (≤10 ng/mL)	6	3

Abbreviations: MMT, manual muscle test; MBI, modified Barthel index; FAC, functional ambulatory category. Values are presented as mean value ± standard deviation.

**Table 2 ijerph-19-08954-t002:** Comparison of changes in absolute BMD after 12 months in each group.

BMD Measures (g/cm²)	Osteoporosis		*p*-Value
	T0	T1	
LS	0.667 ± 0.141	0.722 ± 0.141	0.01 *
FN	0.524 ± 0.073	0.526 ± 0.081	0.74
**BMD Measures** **(g/cm²)**	**Osteopenia**		***p*-Value**
	**T0**	**T1**	
LS	0.855 ± 0.053	0.865 ± 0.061	0.09
FN	0.674 ± 0.091	0.615 ± 0.057	0.01 *

Abbreviations: LS, lumbar spine; FN, femoral neck. Values are presented as mean value (±standard deviation), * *p* < 0.05 was analyzed by paired *t*-test.

**Table 3 ijerph-19-08954-t003:** Correlation between BMD and clinical parameters.

	LS (g/cm²)		FN (g/cm²)	
	r	*p*-Value	r	*p*-Value
MMT of hip and knee	0.05	0.72	0.11	0.49
FAC	0.09	0.44	0.05	0.62
MBI	0.1	0.38	0.06	0.53
OC	−0.16	0.32	−0.19	0.07
CTX	−0.08	0.42	−0.23	0.08
Vit D deficiency	0.18	0.66	0.59	0.12
Vit D insufficiency	0.19	0.12	0.1	0.43
Vit D normal	0.07	0.71	0.19	0.33

Abbreviations: LS, lumbar spine; FN, femoral neck; MMT, manual muscle test; FAC, functional ambulatory category; MBI, modified Barthel index; OC, osteocalcin; CTX, C-telopeptide of collagen type 1; Vit D, vitamin D.

**Table 4 ijerph-19-08954-t004:** Comparison of changes in laboratory finding after 12 months in each group.

Blood Test Level	Osteoporosis		*p*-Value
	T0	T1	
OC (ng/mL)	13.2 ± 5.5	11.36 ± 5.1	0.02 *
CTX (ng/mL)	0.3 ± 0.2	0.21 ± 0.1	0.01 *
Vitamin D (ng/mL)	24.4 ± 10.5	23.1 ± 10.0	0.1
**Blood Test Level**	**Osteopenia**		***p*-Value**
	**T0**	**T1**	
OC (ng/mL)	12.71 ± 6.4	11.18 ± 2.5	0.4
CTX (ng/mL)	0.31 ± 0.2	0.28 ± 0.1	0.7
Vitamin D (ng/mL)	23.6 ± 10.3	22.5 ± 9.9	0.35

Abbreviations: OC, osteocalcin; CTX, C-telopeptide of collagen type 1. Values are presented as mean value (±standard deviation); * *p* < 0.05 was analyzed by paired *t*-test.

## Data Availability

The datasets generated and/or analyzed in the current study are not publicly available but are available from the corresponding author upon reasonable request.

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
