# Peer review of "Change in Bone Mineral Density in Stroke Patients with Osteoporosis or Osteopenia"

_ijerph, 2022, doi:10.3390/ijerph19158954_

Round 1
Reviewer 1 Report
The main aim of the paper „Change of Bone Mineral Density in Stroke Patients with Osteoporosis or Osteopenia is to investigated the presence of osteoporosis or osteopenia in hemiplegic stroke patients, and assessed the need for active rehabilitation treatment by comparing the changes in BMD at a 1-year follow up of osteoporosis patients treated with bisphosphonates and osteopenia patients who received no drug treatment.
The study is interesting. I would like to appretiate the efforts of the authors. However, some facts need to be explained.
Comments:
Abstract:
Line 10: Briefly characterize project participants
Introduction:
I recommend including a mention of whether a similar study existed before.
Line 29: I recommend very briefly write the examples of various samples.
Materials and Methods:
Line 65-66: The list of the comorbidities would appropritable.
Line 68–69: Was the study was conducted in accordance with the updates of the Declaration of Helsinki?
Line 71–72: I would recommend better wording of the sentence.
Line 75: Hematology tests were not conducted in T0?
Line 78: What was the daily time of measurements? Did you measure prior water consumption?
Line 79: In the result section the terms L-spine and femur neck are used. In the discussion cestion the lumbar spine and femoral neck are used. I recemmend unification: lumbar spine (LS) and femoral neck (FN).
Line 93: I recommend including a citation as well as for MBI and MMT. Overall I recommend to more precise description of grouping.
Line 103: I miss the check of normal distribution. Was K-S or S-W test perfomed? Are dat normally distributed? Why there was not used regression analysis to determinate the effect of indicators?
Results:
Line 121: Thre is unnecessary space, before ;
Discussion:
Line 192-193: I miss possible explonation for this finding (increase in lumbar spine (LS) but not in femoral neck (FN). Also there is a lack of osteopenia group LS - FN diff. explanation.
Line 208: I recommend adding this to the conclusion.
Line 220: Other variables influencing higher rick of fall should be mentioned (e.g. muscle strenght, balance ability, ... ). Only decrease in BMD is does not lead to increase risk of falls. More the decrease in BMD lead to increase risk of fracture after the fall.
Line 280: I recommend adding limitation to the possible generalization of the results based on the specific Korean group.
Conclusions:
Line 284: I would say that the possible medication effect should be in conclusion as well.
References:
In the article there is only one reference from the last 5 years, I recommend to expand the number of current references used. There are.
Author Response
Thank you for the paper review.
The points have been revised as you commented.
Line numbers were written based on the revised paper, using ‘the track changes mode in MS Word’.
Also, I uploaded my answer as an attachment.
Point 1: Line 10: Briefly characterize project participants.
Response 1: I appreciate your delicate review. As you commented, we added the contents related to Line 13.
Point 2: I recommend including a mention of whether a similar study existed before.
Response 2: As you commented, we added the contents related to Line 57.
Point 3: Line 29: I recommend very briefly write the examples of various samples.
Response 3: As you commented, we added the contents related to Line 36.
Point 4: Line 65-66: The list of the comorbidities would appropritable.
Response 4: As you commented, we added a list of comorbidities to Lines 77.
Point 5: Line 68–69: Was the study was conducted in accordance with the updates of the Declaration of Helsinki?
Response 5: The study was conducted according to the updates of the Declaration of Helsinki and we added related phrases to Line 85.
Point 6: Line 71–72: I would recommend better wording of the sentence.
Response 6: As you commented, we have revised the contents of Line 89.
Point 7: Line 75: Hematology tests were not conducted in T0?
Response 7: Hematological tests were also performed in T0 and related contents of Line 92 were revised.
Point 8: Line 78: What was the daily time of measurements? Did you measure prior water consumption?
Response 8: We have added content to Line 104 for related matters.
Point 9: Line 79: In the result section the terms L-spine and femur neck are used. In the discussion cestion the lumbar spine and femoral neck are used. I recemmend unification: lumbar spine (LS) and femoral neck (FN).
Response 9: As you recommended, we unified the notation of the entire sentence into Lumbar Spine (LS) and Femoral Neck (FN).
Point 10: Line 93: I recommend including a citation as well as for MBI and MMT. Overall I recommend to more precise description of grouping.
Response 10: As you commented, we added the contents related to Line 121. Additionally, for MBI, Shah et al. (reference [16]) report that MBI scores below 20 points indicate 'total dependency'. Especially in Korea, when performing the disability diagnosis, people who required help entirely in performing daily activities, i.e., those with an MBI score of 32 or lower, are classified as having severe disabilities. Therefore, we divided into two groups based on this and evaluated activities of daily living (ADL).
Point 11: Line 103: I miss the check of normal distribution. Was K-S or S-W test perfomed? Are dat normally distributed? Why there was not used regression analysis to determinate the effect of indicators?
Response 11: In the current study, 63 osteoporosis patients, and 34 osteopenia patients, total of 97 patients were studied. Since the sample size of the two groups is greater than 30, it can be assumed that the variable/amount follows the normal distribution, and because of the central limit theorem, it can be thought that the distribution of these data converges to a normal distribution.
Multiple regression analysis was also performed, but the results were excluded because the meaningful values were not found as in Pearson's correlation analysis.
Point 12: Line 121: Thre is unnecessary space, before ;
Response 12: As you commented, we revised the points you pointed out on Line 154.
Point 13: Line 192-193: I miss possible explonation for this finding (increase in lumbar spine (LS) but not in femoral neck (FN). Also there is a lack of osteopenia group LS - FN diff. explanation.
Response 13: As mentioned above, overall BMD decreases with time and aging of stroke, and especially in stroke patients, bone loss of the hemiplegic side of the femoral neck (FN) is more apparent (reference [6]). Since the osteoporosis group in this study also measured FN BMD on the hemiplegic side, bone loss was expected to appear over time. But it was able to prevent bone loss through bisphosphonate treatment, and it also improved lumbar spine (LS) BMD. In the osteopenia group, because they were not medicated, hemiplegic side FN BMD was decreased as in the general stroke patients. The LS BMD also changed over time, but it seems to be less than in the FN BMD. We added the contents related to Lines 244, and 281.
Point 14: Line 208: I recommend adding this to the conclusion.
Response 14: As you commented, we added related content to the conclusion.
Point 15: Line 220: Other variables influencing higher risk of fall should be mentioned (e.g. muscle strenght, balance ability, ... ). Only decrease in BMD is does not lead to increase risk of falls. More the decrease in BMD lead to increase risk of fracture after the fall.
Response 15: As you commented, we have revised the contents of Line 285.
Point 16: Line 280: I recommend adding limitation to the possible generalization of the results based on the specific Korean group.
Response 16: As you commented, we added the contents related to Line 352.
Point 17: Line 284: I would say that the possible medication effect should be in conclusion as well.
Response 17: As you commented, we added the contents related to Line 359.
Point 18: In the article there is only one reference from the last 5 years, I recommend to expand the number.
Response 18: As you commented, we added the latest reference.
Reviewer 2 Report
Manuscript ID: ijerph-1771601
Title: Change of Bone Mineral Density in Stroke Patients with Osteoporosis or Osteopenia
1.What is the main question addressed by the research?
The aim of this study was to investigate changes in lumbar spine and femoral neck bone mineral density (BMD) in stroke patients with osteoporosis or osteopenia, and to explore the correlation between BMD and osteoporosis-related factors.
2.Is it relevant and interesting?
The article is relevant and interesting.
3.How original is the topic?
The topic is current.
4.What does it add to the subject area compared with other published material?
The authors have collected and analyzed a great deal of interesting data.
5.Is the paper well written?
Yes, the manuscript is well written.
6.Is the text clear and easy to read?
Yes, but minor English editing is required.
7.Are the conclusions consistent with the evidence and arguments presented?
Yes, the conclusions consistent with the evidence and arguments presented but further studies are necessary to confirm authors’ hypothesis.
8.Do they address the main question posed?
Yes, the Authors addressed the main question posed.
Other comments:
- English language: Minor English editing is required.
- Abstract: To attract the reader's attention, please clarify the target of the article, and structure the abstract.
- Study: Specify the type of study (retrosprective?)
- Methods: Which kind of BPs was used in this study? Did you note some adverse events from taking these drugs?
- “Bisphosphonates, one of the drug treatment options, are widely used and their positive effects in 187 stroke patients have been reported in previous studies” I suggest to cite a part about eventual adverse BPs events (MRONJ) refering to recent literature [DOI:10.3390/ph13120423 – DOI:10.1016/j.joms.2020.05.037]
Moreover, I suggest to structure in a better way the graphic order.
After making the indicated changes, the article may be suitable for publication.
Thanks for the opportunity to review this manuscript.
Author Response
Thank you for the paper review.
The points have been revised as you commented.
Line numbers were written based on the revised paper, using ‘the track changes mode in MS Word’.
Also, I uploaded my response as an attachment.
Point 1: English language: Minor English editing is required.
Response 1: Thank you for your advice. We got English editing as you mentioned.
Point 2: To attract the reader's attention, please clarify the target of the article, and structure the abstract.
Response 2: As you commented, we added the phrase in the first part of the abstract.
Point 3: Specify the type of study (retrosprective?)
Response 3: It was prospective study. We added the contents related to Line 60.
Point 4: Which kind of BPs was used in this study? Did you note some adverse events from taking these drugs? “Bisphosphonates, one of the drug treatment options, are widely used and their positive effects in 187 stroke patients have been reported in previous studies”
I suggest to cite a part about eventual adverse BPs events (MRONJ) refering to recent literature [DOI:10.3390/ph13120423 – DOI:10.1016/j.joms.2020.05.037]
Response 4: The bisphosphonate type used in our study was zoledronic acid. As you commented, we added the contents and references about the adverse effects to the method and result.
Point 5: Moreover, I suggest to structure in a better way the graphic order.
Response 5: As you commented, we have changed the Figure placement.

Reviewer 3 Report
Change of Bone Mineral Density in Stroke Patients with Osteoporosis or Osteopenia studied the bone mineral density diversity before and after treatment in stroke patients with Osteoporosis or patients of Osteopenia, respectively. The study found that osteoporosis treatment improved the lumbar spine BMD but decrease femoral neck BMD. The treatment also decreased the BMD of the patients with osteopenia. And BMD Has nothing related with MMT, MBI, and vitamin D of stroke patients with Osteoporosis and patients with Osteopenia.
The correlation between FAC and BMD is overall significant, and it is different before and after the treatment. But the significance in different individual groups is lost when comparing FAC0 and FAC>=1. Does the result indicate the overall effect of the drug treatment in both Osteoporosis and Osteopenia groups? Does it mean the significance may be detected if the author increased the patients’ number? And does it mean the treatment is effective in both Osteoporosis and Osteopenia diseases.
Chi-square can be considered to detect the drug’s effect when considering the increase or decrease of BMD in an individual patient.
Individual data points can be shown in the bar-chart to help readers to understand more about the test.
Author Response
Thank you for the paper review.
The points have been revised as you commented.
Line numbers were written based on the revised paper, using ‘the track changes mode in MS Word’.
Also, I uploaded my response as an attachment.
Point 1: The correlation between FAC and BMD is overall significant, and it is different before and after the treatment. But the significance in different individual groups is lost when comparing FAC 0 and FAC>=1. Does the result indicate the overall effect of the drug treatment in both Osteoporosis and Osteopenia groups? Does it mean the significance may be detected if the author increased the patients’ number?
Response 1: I appreciate your delicate review. What we want to say in the Figure 1 results is to mention the effect of walking training rather than the drug effect. As you can see, there was no statistical difference in each group, but the group capable of walking training (FAC≥1) showed a tendency to improve BMD more. These results are the same as previous studies (reference [28,29]) that more walking training in general stroke patients, regardless of osteoporosis or osteopenia, shows improvement in BMD. If we increase the number of patients, we believe that this statistical significance will be more evident.
Point 2: And does it mean the treatment is effective in both Osteoporosis and Osteopenia diseases.
Response 2: In the case of the osteopenia group, the treatment effect was not known because it was not covered by insurance in Korea and drug treatment could not be performed. However, citing the latest paper (reference [24]), osteopenia women have shown beneficial effects such as increasing BMD and reducing fractures through the treatment of bisphosphonates, we think the osteopenia group will also be effective in drug treatment. (We added the contents related to Line 289). In the future, if it is also covered by insurance in Korea and can be administered in the osteopenia group, we will make an additional research design and compare the differences between the two groups.
Point 3: Chi-square can be considered to detect the drug’s effect when considering the increase or decrease of BMD in an individual patient.
Response 3: Thank you for your careful review. In this study, all osteoporosis groups were medicated, and all osteopenia groups were not medicated due to insurance problems. For the Chi-square test, data is required for those who did not perform drug treatment in the osteoporosis group and those who performed drug treatment in the osteopenia group, but this is not performed, so it will be difficult to apply. In the future, we will divide the presence or absence of treatment for each group and design and analyze it.
Point 4: Individual data points can be shown in the bar-chart to help readers to understand more about the test.
Response 4: I appreciate your delicate review. The reason for choosing the bar chart in this graph is presented to effectively compare the mean of FAC 0 and FAC 1 or more. To provide visualization data that enhances readers' readability, we have shown only mean values, not individual data.

Round 2
Reviewer 2 Report
Authors improved this manuscript, however I strongly suggest to include all the relevant references about BPs related complications.
Author Response
Thank you for the paper review.
The points have been revised as you commented.
Line numbers were written based on the revised paper, using ‘the track changes mode in MS Word’.
Point 1: Authors improved this manuscript, however I strongly suggest to include all the relevant references about BPs related complications.
Response 1: Thank you for your review. As you commented, we added a reference to Line 98.
Reviewer 3 Report
After the revision, the paper's quality is increased. It makes the paper more readable.
Figure 1, FAC 1 or more should be FAC≥1 for the sake of consistency with the main text. The same problem can be seen in Figure 2 A and b.
The label in table 3, the meaning of vit D is unclear, please annotate in the table legend.
Author Response
Thank you for the paper review.
The points have been revised as you commented.
Point 1: After the revision, the paper's quality is increased. It makes the paper more readable. Figure 1, FAC 1 or more should be FAC≥1 for the sake of consistency with the main text. The same problem can be seen in Figure 2 A and b.
Response 1: I appreciate your delicate review. As you commented, Table 1 and Figures 1 and 2 have been modified for the sake of consistency with the main text.
Point 2: The label in table 3, the meaning of vit D is unclear, please annotate in the table legend.
Response 2: As you commented, we have annotated Table 3.